# Optical and Electrophysical Properties of Vinylidene Fluoride/Hexafluoropropylene Ferroelectric Copolymer Films: Effect of Doping with Porphyrin Derivatives

**DOI:** 10.3390/nano13030564

**Published:** 2023-01-30

**Authors:** Valentin V. Kochervinskii, Margaret A. Gradova, Oleg V. Gradov, Andrey I. Sergeev, Anton V. Lobanov, Evgeniya L. Buryanskaya, Tatiana S. Ilina, Dmitry A. Kiselev, Inna A. Malyshkina, Gayane A. Kirakosyan

**Affiliations:** 1Laboratory of Polymer Composite Materials, JSC Scientific Research Institute of Chemical Technology, Moscow 111524, Russia; 2N.N. Semenov Federal Research Center for Chemical Physics, Russian Academy of Sciences, Moscow 119991, Russia; 3Laboratory of Physics of Oxide Ferroelectrics, Department of Materials Science of Semiconductors and Dielectrics, National University of Science and Technology MISIS, Moscow 119049, Russia; 4Faculty of Physics, M.V. Lomonosov Moscow State University, Moscow 119991, Russia; 5Faculty of Fundamental Physical and Chemical Engineering, M.V. Lomonosov Moscow State University, Moscow 119991, Russia; 6Laboratory of Coordination Chemistry of Alkali and Rare Metals, N.S. Kurnakov Institute of General and Inorganic Chemistry RAS, Moscow 119991, Russia; 7A.N. Frumkin Institute of Physical Chemistry and Electrochemistry RAS, Moscow 119071, Russia

**Keywords:** polymers, ferroelectricity, porphyrins, piezoelectricity, conductivity, relaxation

## Abstract

Polymer films doped by different porphyrins, obtained by crystallization from the acetone solutions, differ in absorption and fluorescence spectra, which we attribute to the differences in the structuring and composition of the rotational isomers in the polymer chains. According to the infrared spectroscopy data, the crystallization of the films doped with tetraphenylporphyrin (TPP) proceeds in a mixture of α- and γ-phases with TGTG^−^ and T_3_GT_3_G^−^ conformations, respectively. Three bonds in the planar zigzag conformation ensures the contact of such segments with the active groups of the porphyrin macrocycle, significantly changing its electronic state. Structuring of the films in the presence of TPP leads to an increase in the low-voltage AC-conductivity and the registration of an intense Maxwell-Wagner polarization. An increased conductivity by an order of magnitude in TPP-doped films was also observed at high-voltage polarization. The introduction of TPP during the film formation promotes the displacement of the chemical attachment defects of “head-to-head” type in the monomeric units into the surface. This process is accompanied by a significant increase in the film surface roughness, which was registered by piezo-force microscopy. The latter method also revealed the appearance of hysteresis phenomena during the local piezoelectric coefficient *d*_33_ measurements.

## 1. Introduction

Ferroelectric polymers based on vinylidene fluoride (VDF) and its copolymers are semicrystalline electroactive materials with multifunctional responses, including piezoelectricity, pyroelectricity, and ferroelectricity. Their excellent mechanical properties, chemical stability, and biocompatibility make them suitable materials for producing sensors, actuators, transducers, nanogenerators, etc. [1,2,3]. Ferroelectric polymers are also widely applied for the design of the novel flexible electronic devices for energy storage and conversion [4,5]. In addition to the technical applications, such polymers are also biocompatible, and may be used as various kinds of biosensors [6,7,8,9], implantable materials, and scaffolds for tissue engineering and regenerative medicine [10]. This problem is also of great importance, since ferroelectrics, as a class of materials, are actively used in biomedical research [11]. The values of the acoustic impedances of biological tissues and ferroelectric polymers are close, so the latter may have additional advantages over their inorganic counterparts.

The doping of the ferroelectric polymer matrix with different functional additives provides the expansion of the application area of the resulting composites and smart materials due to the combination of active responses to both components. An example of such materials is dye-doped ferroelectric polymer films with optical and electric response, which have been proposed for application in photovoltaic devices [12]. Among the numerous dyes, porphyrin derivatives are widely used as photosensitizers due to their outstanding photophysical properties. The combination of the intense absorption in the visible spectral range and pronounced fluorescence and photochemical activity, resulting in the photoinduced generation of different reactive oxygen species, makes porphyrin derivatives efficient photovoltaic dyes [13,14,15,16,17], optical sensors, and fluorescent imaging agents [18,19,20,21], and efficient visible light photocatalysts [22,23] and photosensitizers in photodynamic therapy [24,25,26,27,28].

Ferroelectric polymers demonstrate polymorphism at crystallization, where at least four crystallographic modifications can be distinguished [29,30,31]. Three of them seem to be the most interesting because the α-phase has a non-polar unit cell, while the unit cells of the β- and γ-phases have a non-zero dipole moment. From the point of view of medical aspects, this seems to be important since the films crystallized in different modifications demonstrate different effects on the certain cells [8,32,33]. Our previous work has shown that the crystallinity and the composition of the rotational chain isomers in the copolymer films formed from the solutions are controlled by varying the solvent type [34]. The copolymer film formed in the acetone solution has the lowest crystallinity [34] and the highest ratio of T_3_GT_3_G^−^ isomers, so this solvent was used for the introduction of porphyrins into the polymer matrix. During the crystallization, the doping agent is displaced into the amorphous phase, which has an increased free volume, therefore, the effect of two porphyrin derivatives on the structure formation and electric properties of the films can be studied in more detail.

In the present work, we used a VDF copolymer with hexafluoropropylene (HFP) with a number of porphyrin derivatives introduced from the common solvent. The polymers under consideration are biocompatible, so their doping with porphyrins may be interesting for medical applications. In this regard, it is important to follow the mechanisms of interaction between the porphyrins and the polymer matrix, namely, the influence of the dopant molecules on the structure and electrophysical properties of the ferroelectric polymer, as well as the influence of the polymer matrix on the electronic state and photophysical properties of the immobilized dyes.

## 2. Materials and Methods

A statistical copolymer of VDF with HFP (8.3 mol% of HFP) was used to obtain the films [34]. As follows from Table 1, a certain fraction of co-monomers in this copolymer had defective intramolecular attachments.

Porphyrin dyes used to dope the polymer films were tetraphenylporphyrin (TPP) and Zn(II) tetraphenylporphyrin (ZnTPP), purchased from TCI Chemicals and used as received. The structures of the porphyrins used are shown in Figure 1. The solvents used were reagent grade acetone (Ac), ethyl acetate (EtAc), methyl ethyl ketone (MEK), and tetrahydrofuran (THF). Among them, acetone after a long-term storage was also used.

Polymer films doped with porphyrin dyes were obtained by pouring from a common solvent. A porphyrin solution in the same solvent was added to a 5% polymer solution under stirring. The resulting solution was poured into Petri dishes and left in the dark under a lid for several days until the solvent completely evaporated. After complete solvent evaporation, the film can be easily separated from the glass substrate if its thickness is ≥30 microns. In our experiments, the film thickness was maintained at 50 ± 10 microns. The dye concentration in the films obtained was about 0.05%, which made it possible to study them by transmission optical spectroscopy. 

Electric measurements required the deposition of 50 nm thick gold electrodes on the films. Low-voltage measurements of the electric characteristics were performed by Novocontrol Concept 40 broadband dielectric spectrometer (Novocontrol, Montabaur, Germany) in the frequency range 10^−1^–10^7^ Hz. The high-voltage polarization and conductivity were measured at a room temperature using a modified system assembled according to the Sawyer–Tower circuit. 

Optical images of the copolymer films in the transmission mode were obtained using a binocular optical microscope BS-702B with a UCMOS08000KPB digital USB camera (Toup Tek Photonics, Hangzhou, Zhejiang, China), based on a 1/2.5″ CMOS sensor with a resolution of 3264 × 2448 pixels equipped with Altami Studio 3.4.0 software. SEM images of the copolymer film surface were registered by a scanning electron microscope JEOL JSM-T330A (Tokyo, Japan) at the accelerating voltage of 10 kV. The samples were sputter coated with metal (~10 nm Au layer) using the JEOL JFC-1500 vacuum deposition unit. Image registration was performed using a synchronized scheme by the Canon PowerShot A590 IS digital camera (Canon Inc., Ota, Tokyo, Japan).

IR spectra were obtained on Bruker Equinox 55s FTIR spectrometer. Measurements were performed in the transmission and attenuated total reflection (ATR, diamond crystal) regimes (the latter was used to probe the 0.5–2.0 µm thick surface layer of the polymer films). In the ATR method, the principle of attenuated total internal reflection is used. 

Proton relaxation times were measured on a Minispec PC-120 Bruker (Ettlingen, Germany) NMR spectrometer operating at a frequency of 20 MHz. The measurements were carried out at 40 °C using deuterated acetone (Ac-D_6_, Merck, Darmstadt, Germany) as a solvent. The spin-lattice relaxation time (*T*_1_) was determined using a pulse sequence of 180–90° (inversion-recovery) under the following conditions: 90° pulse duration—2.3 μs, *n*—number of points in the decay curve—20, delay between the scans—4–10 s. The spin-spin relaxation time (*T*_2_) was determined using the Carr-Purcell-Meiboom-Gill (CPMG) pulse sequence under the following conditions: 90° pulse duration—2.3 μs, *τ*—time between 90° and 180° pulses—150 μs, *n*—number of points on the decay curve—1500, accumulation of 9 scans, delay between the scans—4–10 s. The obtained curves of the changes in magnetization were analyzed using the discrete multi-exponential decomposition program MULTIT1a and MULTIT2a (Bruker), or the curves were digitized with the frequency of 1 MHz and the ORIGIN 9 software was used. The relative error of the relaxation measurements did not exceed 6%. The diffusion coefficient *D* of protons was determined on a Minispec PC-120 Bruker (Germany) NMR spectrometer with the two-pulse sequence pulsed gradient unit [34]. *D* values were obtained from the Equation (1) describing the dependence of the spin echo relative amplitude on the parameters of the pulsed magnetic field gradient:(1)$$\ln(A_{n}/A_{0})=-\gamma^{2}g^{2}\delta^{2}(\Delta -\delta /3)D,$$
where *A*_n_ and *A*_o_ are the echo amplitudes with and without a pulse gradient, *γ* is the proton gyromagnetic ratio (26.75 × 10^7^ s^−1^ T^−1^), *g* and δ are the amplitude and duration of the pulse gradient, respectively, ∆ is the time between the pulse gradients (time of diffusion), and *D* is the diffusion coefficient. The range of *g* variation is 0.1–2.5 T m^−1^. The variable parameters in the measurements were ∆ and *g*, while *δ* = 500 μs was kept constant. The calibration of the pulse gradient *g* was carried out using the diffusion coefficient of water at 25 °C (*D* = 2.3 × 10^−9^ m^2^ s^−1^). The concentration dependence of the diffusion coefficient *D* for the copolymer in acetone-D_6_ was measured at ∆ = 7500 μs. The TPP was previously dissolved in acetone-D_6_ and then added to the copolymer solution. The solvent content in the sample during diffusion studies was controlled by evacuation (~10^−3^ mm Hg).

The piezoresponse force microscopy (PFM) and local polarization switching spectroscopy measurements were carried out with the MFP-3D (Asylum Research, Goleta, CA, USA) commercial scanning probe microscope using the CSG30/Pt (Tipsnano, Tallinn, Estonia) conductive probe with the spring constant of 0.6 N/m. An alternating current (AC) voltage (10 V, 140 kHz) was superimposed onto a triangular square-stepping wave (*f* = 0.5 Hz, with writing and reading times 25 ms, and bias window up to ±200 V) during the in-field and off-field piezoelectric hysteresis loop measurements (Figure 2). To estimate the effective *d_33_* piezoelectric constants, the deflections and vibration sensitivity of the cantilever alignment were calibrated by the GetReal procedure using the IgorPro software (Asylum Research, Goleta, CA, USA). The on-field and off-field hysteresis loops (PFM Amplitude (pm) and PFM phase) were acquired using the simple harmonic oscillator (SHO) fit with Asylum Research software (Goleta, CA, USA) to exclude the magnification effect of the Q factor of the contact resonance. The effective longitudinal piezoelectric response (“Effective *d_33_*”) was calculated by the Equation: Effective *d_33_* (pm/V) = (PFM Amplitude (pm) × cos(PFM Phase))/Applied AC voltage (V).

The electronic absorption spectra of the solutions and films were recorded using the HACH DR-4000V spectrophotometer in the 320–800 nm wavelength range with 1 nm step. The fluorescence spectra were obtained from the Perkin Elmer LS-50 luminescence spectrometer at the excitation wavelength corresponding to the maximum of the Soret band for the porphyrin used. For solutions, measurements were made according to the standard procedure in 10 mm quartz cells, and in the case of films, at a 90° (for absorption spectra) and 45° (for emission spectra) incidence of the light beam to the film surface fixed on a quartz plate holder.

## 3. Results and Discussion

One of the goals of the study was to find out the effect of the introduced porphyrins on the structure formation and electrophysical properties of the doped polymer films obtained from the solutions. The structure formation in ferroelectric polymer films has been described in more detail in our previous papers [35,36], so here we just briefly note the following circumstances. Since the polymers under study are semicrystalline, the dye molecules will be displaced into the amorphous phase, which has an increased free volume, during the film formation. Therefore, we begin the discussion with the films crystallized from the stored acetone, where, according to our previous data [35,36], the lowest degree of crystallinity is observed. As shown in our recent paper [37], the introduction of other dyes into the polymers under the study leads to additional structuring of the polymer matrix, so we may assume that the introduction of porphyrins into the selected copolymer would lead to a similar result. This is confirmed by the IR spectroscopy data. We have shown previously [35,36] that the film structuring may be judged by the appearance of a doublet at 2850 and 2920 cm^−1^ in the IR spectra (Figure 3). As follows from Table 1, a certain fraction of VDF comonomer both in this [35] and in other copolymers [37] is attached along the chain in the “wrong” position, so that the protons of the C-H groups are far from the fluorine atoms. In this case, the valence vibrations of the mentioned groups are observed in the region of the frequencies where they are usually observed, for example, in polyethylene: 2850 and 2920 cm^−1^. In this regard, the structuring degree can be estimated by the intensity ratio of the bands of antisymmetric vibrations 2920 and 3024 cm^−1^. Examples of such IR spectra for P(VDF-HFP) copolymer films crystallized from acetone are shown in Figure 3. The measurements were performed both in the transmission mode, where the data provide averaged information over the entire sample volume, and in the ATR mode, where a micron-thick surface layer is probed. First of all, the effect of the TPP on the structuring processes was investigated.

Figure 3a demonstrates that the introduction of TPP significantly promotes the structuring of the volume in comparison with the initial film, since the intensities of 2850 and 2920 cm^−1^ bands increased significantly (see Figure 3a and Table 2). The optical micrographs (Figure 4) demonstrate that the structuring which was noted in Figure 3a is accompanied by the formation of micron-sized particles in the volume. 

Table 2 also shows that the introduction of ZnTPP, on the contrary, inhibits the abovementioned structuring processes. At the same time, one should note the increase in the degree of crystallinity (column 3 in Table 2) and the increase in the ratio of isomers in the TGTG^−^ conformation, which can be estimated by the intensity ratio of 530 and 510 cm^−1^ bands [35,36]. A comparison of the transmission and ATR spectra (Figure 3a,b and Table 2) for the TPP-doped film shows that the structuring in the surface is significantly higher than in the bulk. This means that “head-to-head” type defects of the intrachain attachment of neighboring links are intensively forced out into its surface. Accordingly, this should be accompanied by a change in its roughness. This was verified by the scanning electron microscopy (SEM) (Figure 5) and by the PFM study of the surface topography (Figure 6). Emphasis was conducted on revealing differences in the morphology of the surfaces facing the air and the glass substrate. The SEM data qualitatively indicate a significant difference. The quantitative characteristics of such a difference in the roughness were obtained by the PFM method (Figure 6), summarized in Table 3.

It follows from Figure 6 and Table 3 that the surface roughness of the film on the side facing the air is noticeably higher than on the side facing the glass substrate. This phenomenon was previously observed in the films of a VDF copolymer with trifluoroethylene [38] and was associated with the additional influence of moisture adsorbed from the ambient air on the crystallization process of the hydrophobic polymer surface layer.

When TPP is used as a dopant, the structuring in the film volume is also confirmed by the broadband dielectric spectroscopy data, which are shown in Figure 7. In one of the figures, for clarity, the dielectric permittivity components are recalculated in terms of the complex electric modulus *M** [39,40], which components are related to those for the complex dielectric permittivity *ε** (2):(2)$$M^{*}\left(\omega \right)=\frac{1}{\varepsilon^{*}\left(\omega \right)}=M^{\prime }+iM^{\prime \prime }=\frac{\varepsilon ’}{\varepsilon {’}^{2}+\varepsilon ’{’}^{2}}+i\frac{\varepsilon ’’}{\varepsilon {’}^{2}+\varepsilon ’{’}^{2}}.$$

The above curves (especially in the case of TPP doping) indicate the presence of a region of low-frequency dispersion associated with the formation of a space charge *ρ*(*x*,*t*), which in one dimensional case is a function of coordinate *x* (normal to the film surface) and time t. The specified space charge creates a local field *E_sc_*. It can be written as:(3)$$E_{sc}\left(t\right)\approx \int_{0}^{d}\frac{\rho \left(x,t\right)}{\varepsilon \varepsilon_{0}}dx,\;\;0\leq \;x\;\leq d$$

Since the volume charge density and, correspondingly, *E_sc_*, is a function of time, as a result, the dielectric permittivity values at low frequencies (Figure 7a) are above 100, which is not typical of the materials under study [41]. Because the TPP-doped film demonstrates a higher conductivity, the relaxation time for formation of the equilibrium space charge (inset to Figure 7b) is almost two orders of magnitude lower than that for the initial undoped film.

The active role of TPP molecules in the structuring of the polymer matrix is clearly demonstrated in Figure 8, where the frequency dependences of the conductivity and effective dielectric permittivity of acetone and a number of its solutions are presented. In the pure solvent and in its solutions containing a copolymer or TPP, the shape of the conductivity and effective permittivity curves in the low-frequency region indicates the manifestation of electrode polarization. However, in contrast, when the TPP molecules are added to the acetone solution of the polymer, the noted characteristics indicate the formation of a new dispersion region of Maxwell-Wagner polarization. Thus, the TPP induces the polymer structuring even at the stage of the copolymer solution, from which the film is formed.

The mechanism of this unusual phenomenon is associated with the peculiarities of the structure formation of the considered polymers. During their crystallization from a solution, a gel-like state (disappearance of the fluidity of the solution), which is a three-dimensional network formed by intermolecular contacts of adjacent chains or their aggregates, is formed first. The crystals begin to form already in this state [42,43,44]. The real polymers (including the studied copolymer) have a molecular mass distribution. Therefore, the chains of a high molecular weight fraction may form microgel particles in the solution. As applied to our case, the presence of a TPP polymer in the solution may initiate structure formation in such microgels. The system becomes heterogeneous in electric parameters, which may lead to the appearance of Maxwell-Wagner relaxation, which is observed in Figure 8.

A more detailed analysis of the interactions between the porphyrin and the copolymer molecules in acetone during the film formation was carried out using NMR relaxometry. For this purpose, weighed portions of the copolymer and TPP were preliminarily dissolved in deuterated acetone (*Ac*-*D*_6_), and the signal was subsequently recorded during the solvent evaporation. The obtained data on the molecular mobility for the polymer in the presence and in the absence of TPP are shown in Figure 9. The diffusion coefficient values *D* for the pure polymer molecules are higher than those for the polymer samples in the presence of TPP (Figure 9a). This may be due to the change in the shape or size of the polymer particles in the presence of the TPP molecules, which changes the nature of the Brownian motion in the colloid system. The data presented in Figure 9b characterize the molecular mobility as a function of the diffusion time. This dependence confirms the above data on the lower diffusion mobility for the polymer samples in presence of the TPP. A qualitative estimation of the root-mean-square shift <*x*^2^> (according to the Einstein formula, <x^2^> = 6*D*∆, where ∆ is the diffusion time) indicates that the same diffusion time <*x*^2^> for a pure polymer solution is greater than for a polymer with TPP.

The relaxation theory (developed by Bloembergen, Purcell, and Pound) describes the dependence of the NMR relaxation times *T*_1_ and *T*_2_ on the correlation time (*τ*_c_), which characterizes the molecular mobility. According to this theory, the condition *T*_1_ = *T*_2_ characterizes a region with fast motions and short correlation times, for example, a non-viscous fluid. This area is characterized by the values of *τ*_c_ << 1/*ω*_o_, where *ω*_o_ is the measurement operating frequency. The area with *τ*_c_ >> 1/*ω*_o_ is the region of slow motions (for example, in a solid state), where *T*_1_ > *T*_2_. In an intermediate region with *τ*_c_ = 1/*ω*_o_ (for example, in a viscous liquid) *T*_1_ ≥ *T*_2_. Thus, the *T*_1_/*T*_2_ ratio characterizes the changes in the molecular mobility in the system. Figure 9c shows the changes in the *T*_1_/*T*_2_ ratio with the changing polymer concentration with and without TPP in acetone-*D*_6_. It is seen that at the same polymer concentration, the molecular mobility determined by protons for the polymer-dye system is lower than that for a pure polymer solution. It should be noted that the values of *T*_1_ during the concentration measurements varied from ~360 ms (3–5 g of the polymer per 1 g of acetone-*D*_6_) to ~100 μs for a pure polymer solution. The *T*_2_ values varied within the similar limits. Thus, the diffusion and relaxation data indicate a limited mobility in the system of proton-containing polymer molecules in the presence of TPP. This may be due to the chemical interaction between the methylene groups of the copolymer and porphyrin molecules during the film formation which corresponds to the above data.

The kinetics of the space charge formation were studied under high-voltage polarization when a rectangular pulse of the positive polarity field was applied to the sample using the method described earlier [35,36]. The kinetic curves (which are not presented here) were described under the assumption that the films under study are not ideal ferroelectrics, since they may contain impurity ions (e.g., catalyst residues, etc.). Therefore, the above curves were described by the general relationship:(4)$$D=\varepsilon_{0}E+\sigma t^{m}E+2P_{r}\left[1-\exp\left(-{\left(\frac{t}{\tau_{s}}\right)}^{n}\right)\right],$$
where the second term is responsible for the conductivity *σ* contribution, and the third term is responsible for the switching of the bound charges with the characteristic time *τ_s_*. The field dependences of the conductivities obtained from the Equation (4) are shown in Figure 10. For the initial and ZnTPP-doped films, the conductivity increases with the field according to the law, which is qualitatively predicted by the Poole-Frenkel relation (5)
(5)$$j_{PF}=n_{T}\mu qE\cdot \exp\left[-\frac{E_{T_{0}}-\beta_{PF}E^{\frac{1}{2}}}{kT}\right].$$

However, it should be noted, that the value of the characteristic parameter *β*_PF_ is higher in the case of the undoped film, where the structuring during crystallization is more pronounced (see Figure 3 and Table 2). From the same Table and inserts in Figure 10a, one can see that the film crystallization in the presence of ZnTPP promotes the formation of the chain parts in TGTG^−^ conformation typical for the α-phase. It has been previously proposed to estimate the change in the crystallinity by the intensity ratio of the 614 and 600 cm^−1^ bands [35,36]. Table 2 shows that the degree of crystallinity for the films formed in the presence of ZnTPP is higher. Both circumstances, apparently, lead to a decrease in the film conductivity.

As seen from Figure 10b, the TPP-doping of the film leads to two differences. Firstly, its conductivity, all other conditions being equal, is several times higher. Particularly, this may be due to a reduction in crystallinity (see Table 2), but this factor is not decisive. Indeed, since no conductivity increase is observed when the ZnTPP is introduced into the film, it follows that the change of the electronic states in the TPP molecule due to coordination with Zn (Figure 1) plays an important role in the charge transfer processes. The second important difference is that the Poole-Frenkel law is not fulfilled in the TPP-doped films, since at fields higher than some critical one, the conductivity “anomalously” decreases. The insert in Figure 10b shows that such conductivity changes are irreversible. Indeed, after a cycle of high-voltage polarization, the conductivity of the film decreases. It is especially clearly seen in the region of low frequencies, where the relaxation of the space charge is observed (Figure 7c). The noted “anomalous” decrease in the conductivity with the field growth in the polymers under consideration has been previously observed [45] and was associated with the decrease in the carrier concentration due to their capture on the deep traps, which are polar planes of non-centrosymmetric crystals. In our case, the situation is more complicated, since the crystallinity of the films is low. Figure 10a shows that the “anomalous” conductivity decrease in the chosen range of fields is not observed in the initial film. Therefore, its detection in the TPP-doped film should be attributed to the large space charge formed in it. One should remember that transitions from one polymorphic modification to another are possible in ferroelectric polymers under the external electric field [46]. When the degree of crystallinity is low (as in our case [35,36]), part of the chains of the amorphous phase may transform into crystal, especially in the boundary regions of these phases. Thus, the reduction in the amorphous phase ratio should lead to the decrease in conductivity, which is experimentally confirmed by the curves shown in Figure 10b. The role of a large space charge in the TPP-doped copolymer film is that it changes the local field *E_l_*, which can be written as:(6)$$E_{l}=E_{ext}+E_{sc},$$
where *E*_ext_ is the external field.

In this connection, the estimation of the internal field of a ferroelectric polymer becomes very important. The internal field may be controlled by the parameters of the absorption bands of the doping dye [47,48]. The authors emphasize that the internal field is formed both by the bound charges and by the formed space charge. Thus, on the example of the films crystallized from acetone in the presence of the TPP, where the maximum space charge is formed, we can check the validity of the conclusions obtained in [47,48].

Since the solvents chosen for the study differ several times in dielectric permittivity [35], it was first necessary to check whether the nature of the solvent could affect the absorption and fluorescence spectra of porphyrins.

As seen from Figure 11, there is no such dependence in the frequency positions of the absorption and emission bands. On the other hand, a comparison of the absorption spectra of the TPP in the film and in the acetone (Figure 12) shows that the former has a pronounced change in the parameters of the main absorption band. Based on the data obtained, it can be assumed that this change is due to the influence of the internal field of the ferroelectric polymer matrix.

It is seen from Figure 12 that the absorption and emission spectra of the TPP solution in acetone and of the TPP-doped film differ. In particular, the Soret band is batochromically shifted by 20 nm, and the number of Q-bands decreased from four to two. Such changes are typical of the diprotonated form of the TPP formed by the addition of two protons to the nitrogen atoms in the macrocycle core [49,50]. This form is also characterized by the emission band with a peak at 690 nm, clearly observed for the TPP in the polymer film and significantly different from the two-band emission spectrum of the free base TPP in acetone (Figure 12, insert). A gradual transition to the diprotonated form of the TPP as the polymer film crystallizes is confirmed by the data shown in Figure 13.

The series of curves shown in Figure 13a reflect the changes of the dye absorption spectra as the internal field is formed in the ferroelectric polymer matrix. During the first few days after the film formation, it gradually changes its color from pinkish to green, which corresponds to the transition from the neutral to diprotonated form of the TPP. This may be due to a gradual increase in the degree of crystallinity [36] during the first days after the film preparation (at a low crystallization temperature), which causes a redistribution of the space charge regions and changes in the equilibrium internal field of the ferroelectric. The absorption spectrum after four days of the film storage corresponds to the final result of this process. For comparison, Figure 13b shows the spectra of the same film exposed sequentially to the hydrogen chloride and ammonia vapors. It follows from the Figure 13b that the exposure of the freshly prepared film to HCl vapor results in the absorption spectrum similar to that of the film aged for 4 days. In this regard, one can conclude that the improvement of the structure and formation of an equilibrium internal field results in the formation of a diprotonated form of TPP. Since crystallization from acetone produces a large ratio of isomers in the T_3_GT_3_G^−^ conformation [35,36], the presence of three bonds in the planar zigzag conformation should increase the probability of an interaction between the mobile protons of the CH_2_ groups in the copolymer with the donor nitrogen atoms in the TPP macrocycle, and thereby promote its protonation. Obviously, such changes should be observed only for the free base TPP ligand.

The changes in the electronic absorption spectra of the doped films described above disappear if the free base TPP is replaced by its zinc complex ZnTPP, in which all four nitrogen atoms are coordinated by the metal ion and, therefore, do not exhibit acid-base properties. As seen from Figure 14, the frequency position of the main absorption band for ZnTPP in the acetone solution and in the copolymer film is the same. However, it is noticeably broadened, and the extinction is reduced in the case of the film, which indicates the aggregation of the porphyrin molecules in the polymer matrix. During the crystallization, the dye molecules are displaced into the amorphous phase and their local concentration increases. It may be accompanied by the formation of, at least, dimers. This conclusion is confirmed by the significant quenching of ZnTPP fluorescence in the film (Figure 14, insert). The described differences point to the special role of the acid-base properties of the TPP molecule due to the presence of four nitrogen atoms in the macrocycle core in the changing of its electronic structure when introduced into the ferroelectric polymer matrix.

The processes of structure formation during crystallization from the EtAc solutions proceeds with the predominant formation of the α-phase [35]. The films obtained from solutions in MEK and THF also crystallize in the α-phase [35]. As shown in Table 4 (column 2), the films obtained from these three solvents have a high degree of crystallinity. This process leads to a higher degree of structurization, in which the chemical defects of the attachment of the neighboring monomeric units are displaced into the amorphous phase (see column 3 in Table 4).

Thus, significant changes in the electrophysical and optical characteristics of TPP-doped copolymer films are possible only in films with a low degree of crystallinity and in the presence of a large concentration of T_3_GT_3_G^−^ isomers. The possibility of piezoelectricity was tested namely on these films. This was performed by studying polarization switching using local piezoelectric hysteresis loops (see diagram in the experimental section), which are shown in Figure 15. The field of an external source generated by a potential up to 200 V was estimated by relation (6), in which the probe tip was represented as a charged sphere. The distribution of the radial field strength is given by the following expression [51,52]:(7)$$E(r)=\frac{C_{t}U_{t}}{2\pi \varepsilon_{0}\left(\sqrt{\varepsilon_{0}\varepsilon_{a}+1}\right)}\cdot \sqrt{\frac{\varepsilon_{a}}{\varepsilon_{0}}}\cdot \frac{R+\delta }{{\left[{\left(R+\delta \right)}^{2}+r^{2}\right]}^{\frac{3}{2}}},$$
where *r* is the radius of the domain formed, *R* is the radius of the probe tip (30 nm), *δ* is the distance from the tip to the surface, *C_t_* is the capacitance of the system consisting of the tip and the dielectric semi-infinite crystal (1.7·10^−17^ F), and *ε_c_* and *ε_a_* are the relative dielectric permittivity in the polar and non-polar directions, respectively, which we considered the same due to the lack of such data.

As seen from Figure 15a–d, the piezoresponse hysteresis curves differ significantly depending on the measurement scheme. This phenomenon is also observed in inorganic ferroelectrics [53,54,55,56]. Clear hysteresis curves are observed only when a field pulse (with a duration of 25 ms) is followed by the same section in the absence of the field (polarization “without field”) (Figure 2b). This figure shows that the coercive field for the TPP-doped film is lower than that for the undoped film (Figure 15a,b). On the other hand, as discussed above (Figure 3 and Table 2), the addition of the TPP causes a more pronounced structuring during the crystallization. The increased concentration of the surface defects reduces the interaction energy between the neighboring chains in the domains, and therefore lower fields are required for switching. All this may also be reflected in the hysteresis characteristics obtained by the PFM method, since, according to [57], this method also probes the surface layer.

When using the “in-field” scheme (Figure 2a), we observe linear dependences almost without hysteresis (Figure 15c,d). The effective piezoelectric modulus values are very high, which can be useful for the biomedical applications of such polymers. The reason for the difference in the PFM response in the case of the measurement scheme presented in Figure 2a,b must take into account the fact that ferroelectric polymers cannot be classified by structure, such as classical inorganic materials. The difference is that there is always an amorphous phase along with crystals (including those with a noncentrosymmetric lattice) in semicrystalline polymers. The fraction of the amorphous phase in PVDF is ≈0.5 and may reach ≈0.8 in the copolymers under study [35,36]. At a room temperature, the polymer chains in the amorphous phase undergo micro-Brownian cooperative mobility [41]. The relaxation time of such a mobility in our copolymers is ~0.1–1 μs. This follows from Figure 7b, where the high-frequency peak of the loss factor is responsible exactly for this type of mobility [41].

The data presented were obtained at low fields, but the abovementioned segmental mobility should also manifest itself at high fields. This is illustrated in Figure 15e, where the macroscopic dielectric response to a positive field pulse of 16 MV/m is shown for the initial film. It is seen that when the field is turned on, a fast response occurs, which we associate exactly with the response to the dipoles in the amorphous phase to the field. After the field is turned off, there is again a sharp decrease in *D*, which should be attributed to the disappearance of the contribution of the above mobility to polarization. The subsequent curve is responsible for the remanent polarization relaxation.

The scheme in Figure 2b in the PFM method describes such a situation. The duration of the read (as well as write) period is 25 ms, which is several orders of magnitude higher than 1 μs. During this time, all the dipoles of the amorphous phase reach the equilibrium state, and the space charge formed in the ferroelectric can partially compensate the polarization of the bound charges of the existing polar crystals. Such processes may be the cause of the hysteresis observed in the experiment, which is shown in Figure 15a,b. The scheme in Figure 2a excludes relaxation processes, and the contribution of the amorphous phase chains to the observed polarization will be constant as the field increases. In fact, we should observe the contribution to the piezoresponse from electrostriction [29,30,58,59,60], which, according to the literature, is due to the amorphous phase [61]. A comparison of the curves in Figure 15c,d shows that the introduction of TPP into the copolymer strongly increases the maximal piezoeffect. Since the TPP molecules are displaced into the amorphous phase during crystallization, the named molecules increase the contribution to the piezoresponse from the amorphous phase. The mechanism of this effect may be related to the fact that the space charge field created by the TPP molecules (located in the amorphous phase) can lead to the conformational transitions of the T_3_GT_3_G^−^ → (-TT-)*_n_* type, where *n* ≥ 4. The long sequences formed in this case in the planar zigzag conformation have a higher dipole moment, and therefore the piezoresponse should increase, which is observed when comparing Figure 15c,d.

An important circumstance related to the biomedical application of ferroelectric polymers should be noted. Since the introduction of TPP into the film leads to an increase in the conductivity, which can be controlled by the field, there is potentially a new area of application of such polymers in practical biomedicine. While previously the perspectives of their use in regulating the characteristics of biocompatible materials through the bound charges were discussed [10], now there are new perspectives of their use by regulating their properties related to the presence of the (quasi-free) carriers.

## 4. Conclusions

The effect of the internal field of vinylidene fluoride-based ferroelectric copolymer films doped with porphyrin molecules on their optical and electrophysical properties has been studied. For the films with low crystallinity, the introduction of tetraphenylporphyrin molecules into the polymer matrix causes intensive structuring related to the displacement of the defects into the amorphous phase, and especially to the surface. The introduction of porphyrins into the polymer matrix is accompanied by a strong luminescence quenching with respect to the solution. One of the reasons for this is the increase in the probability of the quenching of the excited electronic states due to the interaction between the neighboring dye molecules within the amorphous phase. However, one cannot exclude their interaction with the atoms of the polymer matrix. This is indirectly evidenced by the significant increase in the conductivity of the polymer matrix doped with the tetraphenylporphyrin molecules. The ferroelectric properties of P(VDF-HFP) were studied by piezoforce microscopy. The structuring processes are accompanied by a noticeable increase in the surface roughness of the films. The study of ferroelectric hysteresis shows that it is noticeably pronounced only when using the “out-of-field” measurement scheme. For the “in-field” measurement scheme, the hysteresis is practically not detected, but the values of the local piezoelectric coefficient *d*_33_ reach 400 pm/V. An explanation of this phenomenon is given based on the two-phase model of the crystalline polymer, where the fraction of the amorphous phase may reach 0.8. The obtained hybrid organic materials with an optical response can be used as colorimetric pH sensors and components of photovoltaic devices. The properly chosen conditions for obtaining such materials allows combining both the valuable electrophysical properties of the polymer and the optical properties of the dopant, thereby providing an active multifactorial response to the resulting smart materials.

## Figures and Tables

**Figure 1 nanomaterials-13-00564-f001:**
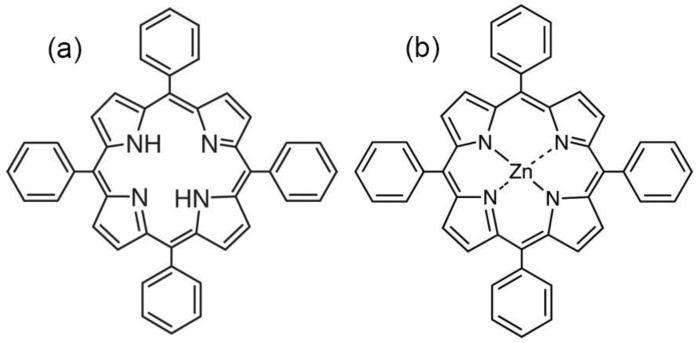
The structure of TPP (**a**) and ZnTPP (**b**).

**Figure 2 nanomaterials-13-00564-f002:**
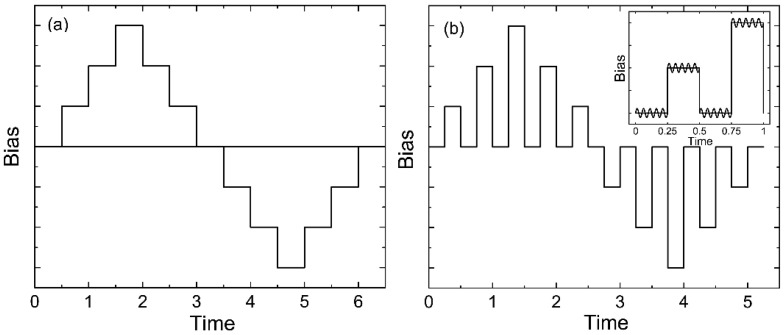
Schemes for obtaining hysteresis loops in two modes—in-field (**a**) and off-field (**b**).

**Figure 3 nanomaterials-13-00564-f003:**
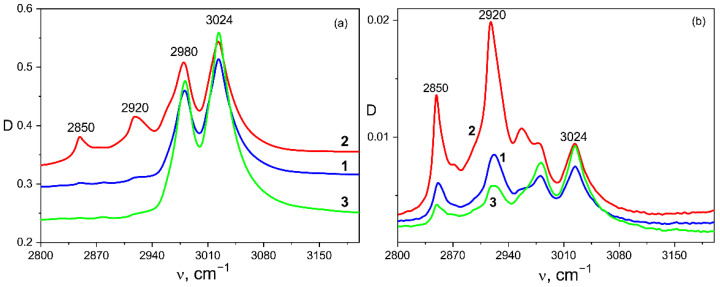
Vibrational bands in the valence vibration region of C–H groups for P(VDF-HFP) copolymer films crystallized from acetone without doping 1—and with different dopants: 2—TPP, 3—ZnTPP; (**a**) transmission mode, (**b**) ATR mode.

**Figure 4 nanomaterials-13-00564-f004:**
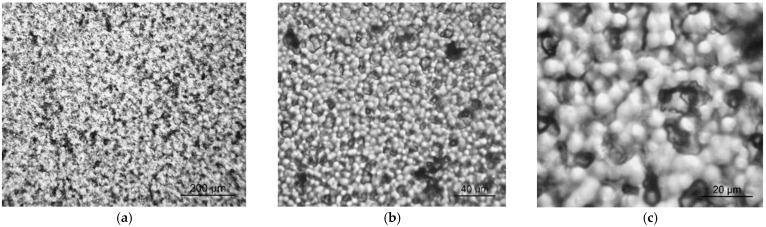
Optical micrographs of TPP-doped P(VDF-HFP) film (magnification ×100 (**a**), ×400 (**b**) and ×1000 (**c**)).

**Figure 5 nanomaterials-13-00564-f005:**
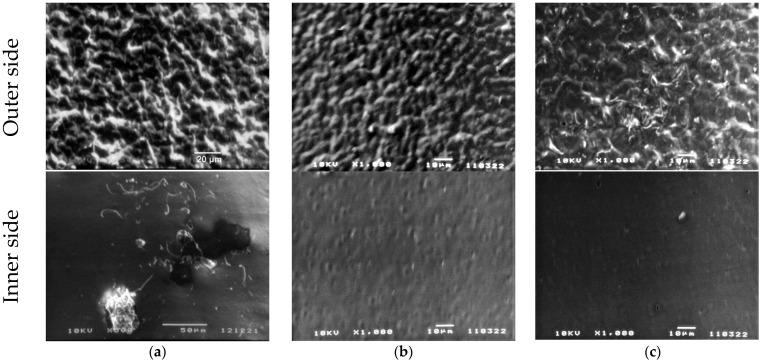
SEM images of P(VDF-HFP) films: (**a**) initial, (**b**) TPP-doped, (**c**) ZnTPP-doped.

**Figure 6 nanomaterials-13-00564-f006:**
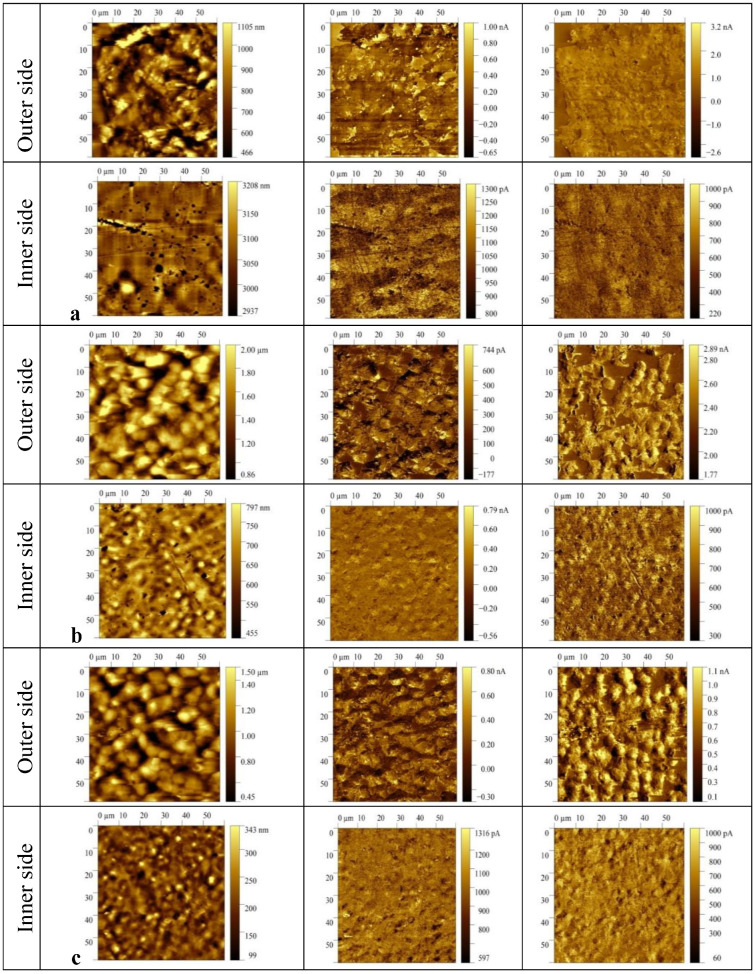
PFM images of the film surface topography (left), vertical (middle), and lateral (right) piezoresponse signals for P(VDF-HFP) films: (**a**) initial, (**b**) TPP-doped, (**c**) ZnTPP-doped.

**Figure 7 nanomaterials-13-00564-f007:**
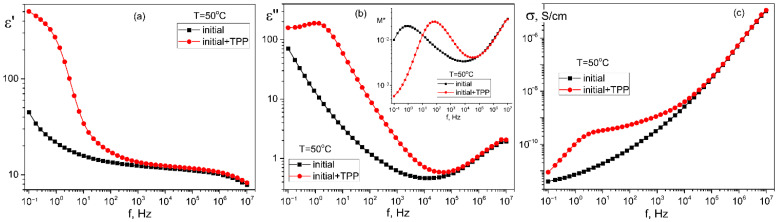
Comparison of the frequency dependences of the real *ε*′(f) (**a**) and imaginary *ε*″(f) (**b**) components of the complex dielectric permittivity and the real part of the AC-conductivity σ(f) (**c**) in the initial and TPP-doped P(VDF-HFP) copolymer films crystallized from acetone; in the insert in Figure 7b—frequency dependences of the imaginary part of the electric modulus M″(f). T = 50 °C.

**Figure 8 nanomaterials-13-00564-f008:**
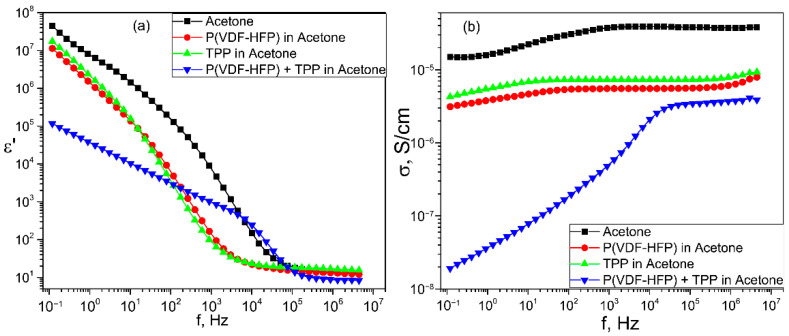
Frequency dependences of dielectric permittivity *ε*′(f) (**a**) and conductivity *σ*(f) (**b**) of acetone and in its solutions with TPP, pure copolymer and of mixed solution of P(VDF-HFP) and TPP.

**Figure 9 nanomaterials-13-00564-f009:**
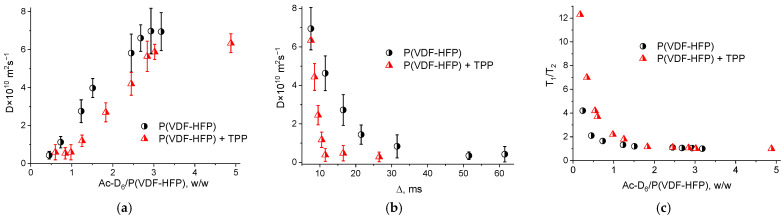
Diffusion coefficient dependence on the solvent/polymer mass ratio (**a**) and on the diffusion time (**b**); dependence of the T_1_/T_2_ ratio on the solvent/polymer mass ratio (**c**).

**Figure 10 nanomaterials-13-00564-f010:**
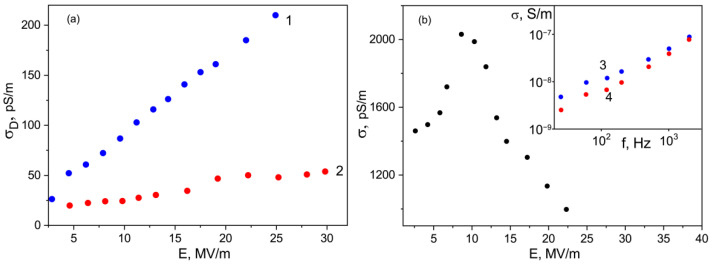
Field dependences of conductivities under a rectangular pulse of the electric field of the initial ((**a**), 1), ZnTPP- ((**a**), 2) and TPP-doped copolymer films (**b**) obtained from acetone; insert in Figure 10b: low-voltage conductivities of the films before (3) and after (4) high-voltage polarization.

**Figure 11 nanomaterials-13-00564-f011:**
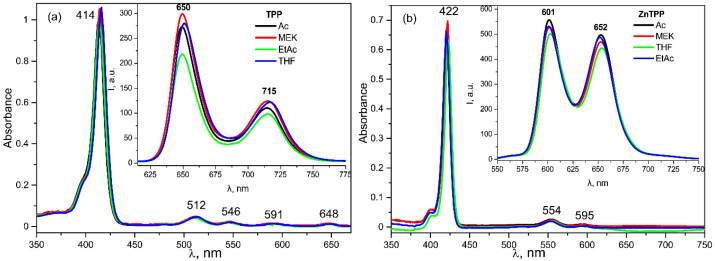
Electronic absorption and fluorescence (insert) spectra for TPP (**a**) and ZnTPP (**b**) in different solvents.

**Figure 12 nanomaterials-13-00564-f012:**
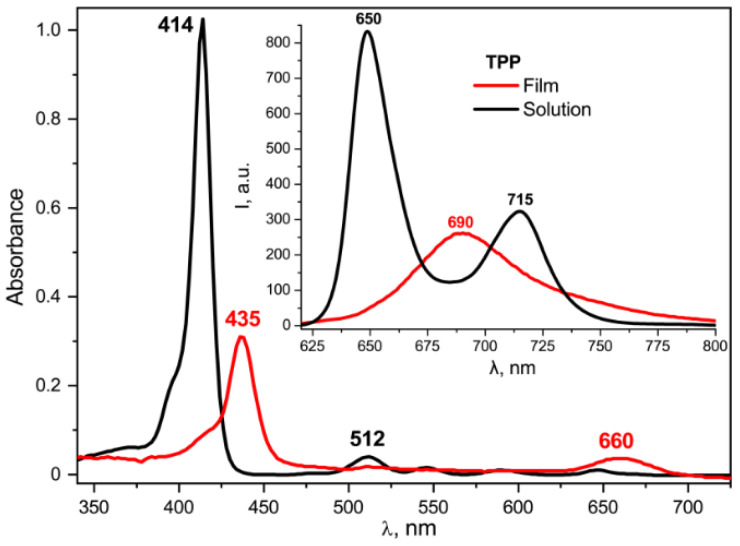
Comparison of the absorption and fluorescence (insert) spectra for TPP in the acetone solution and in the copolymer film crystallized from acetone.

**Figure 13 nanomaterials-13-00564-f013:**
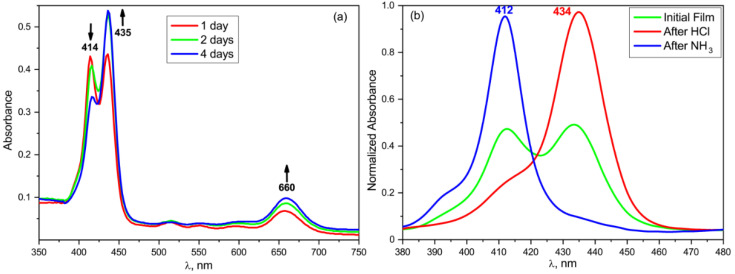
Changes in the absorption spectra of the TPP-doped film with time during storage (**a**) and during sequential exposure to ammonia and hydrogen chloride vapor (**b**).

**Figure 14 nanomaterials-13-00564-f014:**
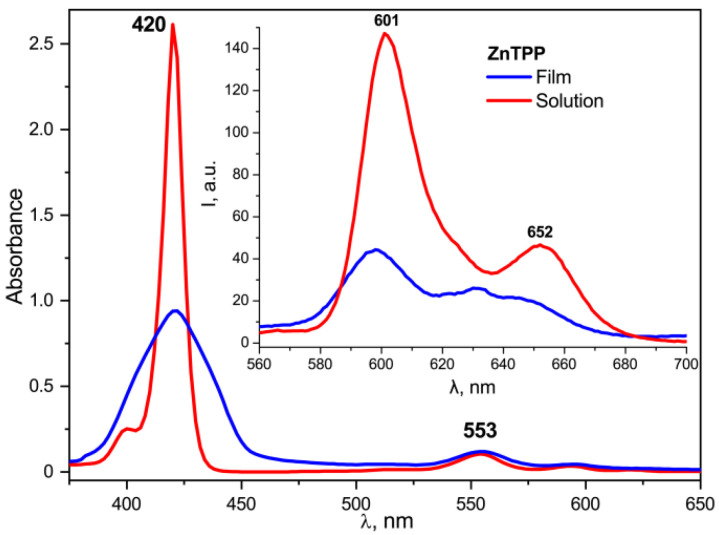
Comparison of the absorption and fluorescence (insert) spectra for ZnTPP in the acetone solution and in the copolymer film crystallized from acetone.

**Figure 15 nanomaterials-13-00564-f015:**
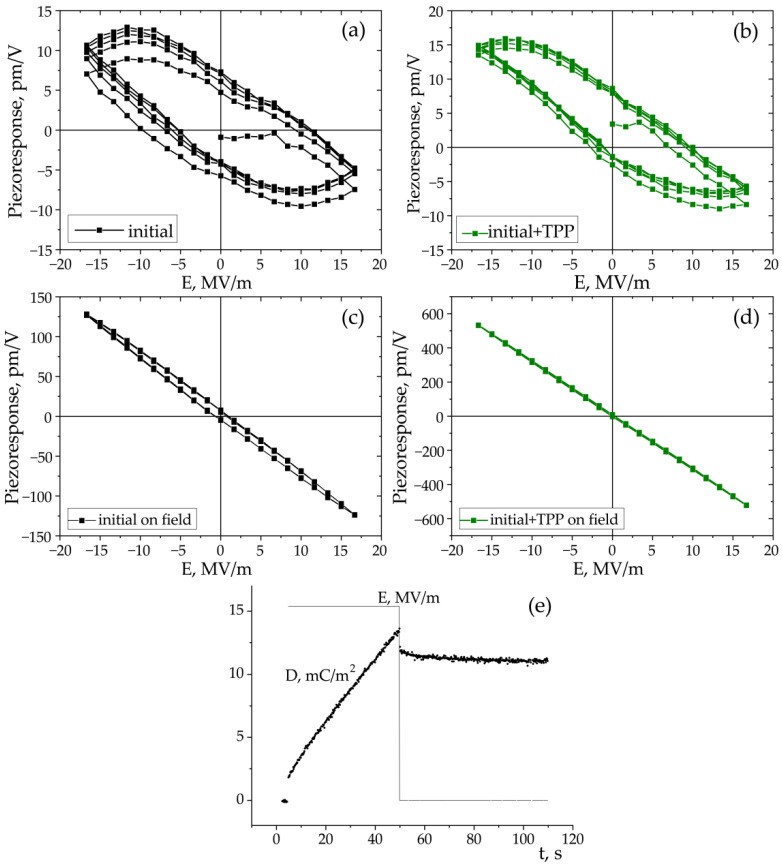
Local hysteresis loops obtained in “out-of-field” (**a**,**b**) and “in-field” (**c**,**d**) modes; (**e**) response of the initial film to a rectangular field pulse followed by the signal relaxation.

**Table 1 nanomaterials-13-00564-t001:** Composition of the P(VDF-HFP) copolymer established by NMR ^19^F data [34].

Monomer	Mol. Fraction
Normal VDF	88.3
Reverse VDF	3.4
Normal HFP	7.8
Reverse HFP	0.5

**Table 2 nanomaterials-13-00564-t002:** Intensity ratio of a number of bands in the IR spectra of doped P(VDF-HFP) copolymer films crystallized from acetone.

Sample	Absorption D_530_/D_510_	Absorption D_614_/D_600_	Absorption D_2920_/D_3024_	ATR D_2920_/D_3024_
Initial film	0.40	0.68	0.05	1.2
TPP-doped film	-	0.57	0.4	2.0
ZnTPP-doped film	0.53	1.24	0	0.5

**Table 3 nanomaterials-13-00564-t003:** RMS roughness values (nm) of the films crystallized from the acetone solutions.

Sample	Inner Side	Outer Side
Initial film	48	170
TPP-doped film	58	284
ZnTPP-doped film	35	227

**Table 4 nanomaterials-13-00564-t004:** Intensity ratios of the characteristic bands in IR transmission spectra for copolymer films crystallized from different solvents.

Solvent	D_614_/D_600_	D_2920_/D_3024_
Ac	0.9	0.06
MEK	4.6	0.17
THF	4.8	4.0
EtAc	4.7	0.18

## Data Availability

The data presented in this study are available on request from the corresponding author.

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
