# Peer review of "Optical and Electrophysical Properties of Vinylidene Fluoride/Hexafluoropropylene Ferroelectric Copolymer Films: Effect of Doping with Porphyrin Derivatives"

_nanomaterials, 2023, doi:10.3390/nano13030564_

Round 1

Reviewer 1 Report

In this work, the authors discuss the optical and electrophysical properties of copolymer films, which is affected by doping porphyrin derivatives. Some interesting physical phenomena are found and discussed. However, some fundamental problems still need to be clarified.

(1)    The structural stability about local hysteresis loops need to confirmed, at least some cyclic tests.

(2)    The absorption spectra may be affected by the TPP-doped concentration. The authors can provide some PL evidences.

(3)    What is the physical mechanism for the absorption spectra change as during time? The structural changes or others? What are the evidences?

(4)    When the solution is changed into film, the absorbance behaviors are changed. If the films are changed into solution, what will happen?

(5)    The time-related PL decay curves and PLE results can better understand your findings.  

Reviewer 2 Report

1.       Please check whether the topic of Figure 10 is correct? Figure 10 does not have a and b.

2.       Why choose ZnTPP? Why not consider other metals TPP? Whether different metals affect the performance.

3.       Comparing Figure 8 with Figure 10, why the absorption spectra and fluorescence in TPP film and solution are quite different, while ZnTPP does not change much.

4.        It is recommended to supplement photos and scanning electron microscopy of film in the text.

5.       Consider putting the experimental diagram into the main text.

Reviewer 3 Report

please see the comments as following:

1. The author is suggested to explain the full name of 'ATR mode' as not every audience is familiar with it. 

2. The author was suggested to add AFM surface pictures to show the roughness. (together with table 3) Also, how to get the roughness of "internal side" in Table 3if it is contacted the substrate?

3. As the author indicates at Line 209 that dielectric constant cannot be 100 for the materials in this work, so the author is suggested to explain more why the discrepancy happened. 

4. The author is suggested to add cartons to describe the thin film structures used to get the P-E hysteresis loop. Which substrate is it? is there electrodes? Also, the author is suggested to add the P-E loops of doped and un-doped polymer comparison. 

5. Figure 11e and Figure 2b have the same content. 

6. It is not clear on the significance of the work, although comprehensive optical study has been done on the doped co-polymers. In another word, by doping, which property has been improved or which new phenomenon has been observed? Maybe the author can re-organize the language in abstract or conclusion , so that the importance of this work can be emphasized 

7. Figure 10, there is no 'a' and 'b' in Figure 10. 

Round 2

Reviewer 1 Report

Can be accepted in present form

Author Response

Author's Response to Reviewer 1

We thank the Reviewer for the comments on the study. The changes have been made to the manuscript (highlighted in yellow).

Reviewer 3 Report

Thanks the author for the replying and the new version still needs some minor edits that Line 486, the content is not English. 

Author Response

Author's Response to Reviewer 2

We thank the Reviewer for the comments on the study. Based on the useful comments, the changes have been made to the manuscript (highlighted in yellow).

1) English language and style are fine/minor spell check required

Author Response:

The English spelling was corrected

2) Thanks the author for the replying and the new version still needs some minor edits that Line 486, the content is not English.

Author Response:

Thanks for the comment; the content was translated into English.